# Estimation of Parameters of Biomass State of Sowing Spring Wheat

Ilya Mikhayilovich Mikhailenko

FGBNU Agrophysical Research Institute, Grazhdansky Prospect, 14, 195220 St. Petersburg, Russia;
ilya.mihailenko@yandex.ru

**Abstract:** The purpose of this work is to present a new method for estimating the parameters of the biomass of agricultural crops based on Earth remote sensing (ERS) data. The method includes mathematical models and algorithms estimation and has been tested on the example of spring wheat sowing. Sowing biomass parameters are the basis for making management decisions aimed at obtaining a given crop yield. Currently, for these purposes, vegetation indices are most widely used. It is impossible to estimate the physical parameters of the crop sowing biomass using these indices, due to their scalar form and lack of dimension. The paper develops a classical approach to the problem of estimating the parameters of the state of agricultural crops, in which remote sensing data are considered as an indirect measurement of the estimated parameters. The basis for the implementation of the estimation method is the dynamic model of biomass parameters and the remote sensing model, which reflects the relationship between the spectral reflection parameters and the estimated parameters of the crop biomass. The parameters of the dynamic model and the remote sensing model are refined by selective ground measurements in separate elementary sections of the field. The difference between this article and previous works of a similar nature lies in the fact that agricultural crops with a more complex morphological structure are considered as the object of evaluation. In addition, such an important feature of agricultural objects as their spatial distribution is considered here. To take it into account, a new type of mathematical models is used, in which spatial coordinates are introduced. Due to the significant complication of modeling and estimation algorithms based on such models, simpler approximation schemes are proposed. The advantage of the proposed approach is that the assessment is considered as a dynamic process that meets the content of the task of monitoring crops.

**Keywords:** remote sensing of the Earth; biomass parameters; crops; parameter estimation; mathematical models; algorithms; spatial correctors

## 1. Introduction

Today, agriculture is being transformed under the influence of bio- and nanotechnologies, varieties, and breeds are being improved by genomics, and manufacturers are moving from a product to a service model, integrating production-sales chains and adapting their products to the needs of a particular consumer. In each of these trends, digital technologies play an important role. However, the penetration of these technologies into the agricultural sector is not deep, and so far, commodity producers pay little attention to modern information technologies (IT). Among such technologies, the use of Earth remote sensing (ERS) data in systems for monitoring the state of crops and soil cover is extremely important. The information of monitoring systems allows solving such typical tasks as providing current control of the condition of crops; preliminary forecasting of crop yields; monitoring the pace of harvesting at the same time in the territories of large regions; determination of the capacity of pastures of various types and the productivity of hayfields. In general, all this makes it possible to implement effective support for management decisions in agriculture [1].

When using remote sensing data in agriculture, it has already become customary to solve most of the above tasks based on special image interpretation technologies. They are obtained by systematic repeated surveys from various devices, which provide observation of the dynamics of the development of agricultural crops and forecasting yields. In this case, various variants of vegetation indices (VI) are most often built, among which the NDVI index (Normalized Difference Vegetation Index) is most widely used. Using such VI, it is possible to judge their agrotechnical state by the color tone of the image of the fields [1–7].

The calculation of most of the vegetation indices is based on the two most stable sections of the curve of the spectral reflectance of plants. This approach makes it possible to obtain only generalized assessments of the state of crops. It is important to note that when calculating any VI, the information potential of the remote sensing method decreases, since any VI is a kind of convolution of signals into a scalar value. This always leads to a decrease in the overall information content of individual channels. To increase the information content of the estimation problem, it is necessary not to decrease, but, on the contrary, to increase the number of independent measurement channels [8]. So, the number of productive sowing indicators that need to be evaluated can reach 10. It is impossible and incorrect to estimate these states by one or several scalar VIs, and it is incorrect according to the scientific statement.

In a large review paper [9], a literature between 2000 and 2019 was conducted, which was devoted to the application of remote sensing technologies in production agriculture, from field preparation, planting, and seasonal application to harvest. The authors found an increasing trend in the use of remote sensing technologies in agricultural production over the past 20 years, with a marked increase in the use of unmanned aerial vehicles (UAVs) after 2015. It is emphasized that remote sensing technologies can be used to support specific management decisions at various stages of crop production, helping to optimize technologies.

The paper analyzes the use of various remote sensing platforms, including portable, aviation, and satellite ones, which can be used to collect data with different spatial, temporal, and spectral resolutions. Various types of sensors are being investigated, including visible (visual), multispectral, hyperspectral, thermal, and microwave.

Particular attention is paid to the possibility of estimating the yield of agricultural crops according to remote sensing data. It is indicated that for these purposes, in addition to remote sensing data, other auxiliary variables, such as weather (for example, solar radiation, temperature, precipitation), vegetation conditions, and soil properties, are supposed to be introduced into consideration using empirical or mechanistic approaches. Because empirical approaches directly relate inputs to outputs through purely statistical means, they are relatively simple, and more data is needed to improve model robustness. On the other hand, mechanistic models focus on causal relationships between inputs and outputs by accounting for the various biophysical processes involved. They often rely on various assumptions that may not always work.

Of particular importance is, as the authors point out, that traditional approaches forthe determination of seasonal crop stresses rely on field survey of crops or laboratory experiments, which are laborious and expensive if extended over large areas. The use of remote sensing provides a timely and non-destructive approach to identifying, quantifying, and mapping crop-related stresses and is thus useful in guiding specific management decisions on nutrient and insecticide application rather than the entire field.

The authors found that in order to improve yield forecasting, previous studies proposed the idea of assimilation of remote sensing data into a mechanistic crop growth model based on seasonal weather forecasts and field management practices. This idea has been around for a long time, but it is still not well understood. The authors argue that the increased availability of more remote sensing data and the development of modern data analysis tools will move research towards the development of crop yield prediction systems.

Despite the very large volume of analyzed sources, the authors did not touch upon the scientific and methodological foundations of the problem of estimating the parameters of the state of agricultural crops based on remote sensing data. At the same time, it can be argued that this work is a prologue to solving such a problem, to which this work is dedicated. The approach developed in it has already been considered in [8,10,11]. It is based on the classical estimation of parameters of the state of agricultural crops according to remote sensing data, considered as an indirect measurement of the state of the object of assessment [12]. This approach has been tested on various forage crops, the biomass of which is the raw material for the preparation of feed. The purpose of this work is to develop a classical approach to assessing the parameters of the state of crops that are more complex in their morphological structure of grain crops.

## 2. Materials and Methods

### 2.1. Mathematical Models

The task of assessing the parameters of the state of the biomass of spring wheat sowing is to construct in real time estimates of such physical parameters as the density of the total biomass and its marketable part (yield), as well as its composition in terms of dry and wet weight. Estimates of these parameters can be used to solve problems of agricultural technology management. The classical approach to the estimation problem is to refine the a priori information about the estimated parameters by the measured parameters, which in our case are the remote sensing data, which are a source of a posteriori information about the state parameters [13–15].

All a priori information about the estimated parameters is contained in mathematical models that reflect the dependence of the parameters on the main influencing factors. Due to the fact that the sowing of spring wheat can have two different biomass structures, before heading and after its onset, two mathematical models are used in the estimation problem [8,16]. Moreover, both models can be represented in a single vector-matrix canonical form:

- in the time interval before the earing of crops

$$\dot{X}_m(y,h) = A_m X_m(t,y,h) + B_m V(t,y,h) + C_m F(t),$$
$$t \in (T_{1m}, T_{2m}), \quad X_m(T_{1m}, y, h) = 0.$$
(1)

- in the time interval from the beginning of earing to the full ripening of the grain

$$\dot{X}_u(y,h) = A_u X_u(t,y,h) + B_u V(t,y,h) + C_u F(t),$$
$$t \in (T_{1u}, T_{2u}), \quad X_u(T_{1u}, y, h) = X_{u0}(y, h).$$
(2)

In models (1) and (2) the following designations are accepted:

$$A_m = \begin{bmatrix} a_{11} & a_{12} \\ a_{21} & a_{22} \end{bmatrix}_m, A_u = \begin{bmatrix} a_{11} & a_{12} & a_{13} \\ a_{21} & a_{22} & a_{23} \\ a_{31} & a_{32} & a_{33} \end{bmatrix}_u$$ —dynamic matrices of models (1) and (2);

$$B_m = \begin{bmatrix} b_{11} & b_{12} & b_{13} & b_{14} & b_{15} \\ b_{21} & b_{22} & b_{23} & b_{24} & b_{25} \end{bmatrix}_m, B_u = \begin{bmatrix} b_{11} & b_{12} & b_{13} & b_{14} & b_{15} \\ b_{21} & b_{22} & b_{23} & b_{24} & b_{25} \\ b_{31} & b_{32} & b_{33} & b_{34} & b_{35} \end{bmatrix}_u$$ —

transfer matrices model controls (1) and (2);

$$C_m = \begin{bmatrix} c_{11} & c_{12} & c_{13} \\ c_{21} & c_{22} & c_{23} \end{bmatrix}_m, C_u = \begin{bmatrix} c_{11} & c_{12} & c_{13} \\ c_{21} & c_{22} & c_{23} \\ c_{31} & c_{32} & c_{33} \end{bmatrix}_u$$ —transmission matrices of external

disturbances of models (1) and (2);

$$X_m(t,y,h) = \begin{bmatrix} x_{1m}(t,y,h) \\ x_{2m}(t,y,h) \end{bmatrix}, X_u(t,y,h) = \begin{bmatrix} x_{1u}(t,y,h) \\ x_{2u}(t,y,h) \\ x_{3u}(t,y,h) \end{bmatrix}$$ —vectors of estimated crop

biomass parameters in models (1) and (2)

$$F(t) = \begin{bmatrix} f_1(t) \\ f_2(t) \\ f_3(t) \end{bmatrix}$$ —vector of external climatic disturbances in models (1) and (2);

$$V(t,y,h) = \begin{bmatrix} v_N(t,y,h) \\ v_K(t,y,h) \\ v_P(t,y,h) \\ v_{Mg}(t,y,h) \\ v_5(t,y,h) \end{bmatrix} \text{—control vector in models (1) and (2).}$$

The biomass state parameters in the model (1) are: $x_{1m}$—the average density of the total biomass of crops over the area of the field, cwt·ha$^{-1}$; $x_{2m}$—the average density of the wet weight of crops over the area of the field, cwt·ha$^{-1}$; the parameters of the state of the biomass in the model (12) are: $x_{1u}$—the average density of the total biomass over the field area, cwt·ha$^{-1}$; $x_{2u}$—the average density of the wet weight of crops over the area of the field, cwt·ha$^{-1}$; $x_{3u}$—the average density of the mass of ears of crops (harvest) over the area of the field, cwt·ha$^{-1}$; external perturbations in both models are: $f_1$—average daily air temperature, °C; $f_2$—average daily radiation level, W·(m$^2$·h)$^{-1}$; $f_3$—average daily precipitation intensity, mm; parameters of the chemical state of the soil: $v_N$—nitrogen content in the soil, kg·ha$^{-1}$; $v_K$ is the potassium content in the soil, kg·ha$^{-1}$; $v_P$—phosphorus content in the soil, kg·ha$^{-1}$; $v_{Mg}$—content of magnesium in the soil, kg·ha$^{-1}$; $v_4$—moisture content in the soil, mm; $y,h$—spatial coordinates, m; $t \in (T_{1m}, T_{2m})$—time variable, days, beginning and end of the growing season preceding the heading phase; $t \in (T_{1u}, T_{2u})$—the beginning and end of the growing season from the beginning of earing to the full ripening of the grain, cwt—center (hundredweight)—the unit of mass adopted in Russia is 1 cwt = 0.1 tn.

The total crop biomass includes the mass of stems and ears, the wet mass is the mass of moisture in the composition of wheat plants, the mass of ears includes the mass of grain and chaff.

When using models (1)–(4), the assumption is made that the elementary areas of the field with spatial coordinates (y, h), into which the field area is divided during estimation, have the same dynamic properties and they do not affect each other. This assumption is correct for leveled reliefs, and this allows oneto significantly simplify the shape of the models.

A priori information about the parameters of the state of the crop biomass, formed by models (1) and (2), must be corrected according to real measurements, for which remote sensing models are introduced. On both time intervals, a one vector-matrix canonical form of the models is used:

- in the time interval before the earing of crops

$$Z_m(y,h) = P_m W(X_m(y,h)) \tag{3}$$

- in the time interval from the beginning of earing to the full ripening of the grain

$$Z_u(y,h) = P_u W(X_u(y,h)) \tag{4}$$

In models (3) and (4) the following designations are accepted:

$\begin{bmatrix} z_{1m}(y,h) \\ z_{2m}(y,h) \end{bmatrix}$—vector of reflection parameters for the spatial coordinate $(y,h)$. in the visible range (400–700 nm) ($z_{1m}$) and in the near infrared range (750–950 nm) ($z_{2m}$);

$\begin{bmatrix} z_{1u}(y,h) \\ z_{2u}(y,h) \\ z_{3u}(y,h) \end{bmatrix}$—vector of integrated reflection parameters in green (500–565 nm)—($z_{1u}$), in red (625–740 nm)—($z_{2u}$), in near-IR (750–950 nm)—($z_{3u}$);

$P_m = \begin{bmatrix} p_{01} & p_{11} & p_{12} & p_{13} & p_{14} & p_{15} & p_{16} \\ p_{02} & p_{21} & p_{22} & p_{23} & p_{24} & p_{25} & p_{26} \end{bmatrix}_m$—model parameter matrix (3);

$W(X_m(y,h)) = \begin{bmatrix} 1 & x_{1m}(y,h) & x_{2m}(y,h) & x_{1m}^2(y,h) & x_{2m}^2(y,h) & x_{1m}^3(y,h) & x_{2m}^3(y,h) \end{bmatrix}$—vector-function, where the arguments are the parameters of the state of sowing: $x_{1m}$—the density of the sowing biomass (yield) for the spatial coordinate $(y,h)$, cwt·ha$^{-1}$; $x_{2m}$—the density of the sowing wet weight for the spatial coordinate $(y,h)$, cwt·ha$^{-1}$;

$$\mathbf{P}_u = \begin{bmatrix} p_{01} & p_{11} & p_{12} & p_{13} & p_{14} & p_{15} & p_{16} & p_{17} & p_{18} & p_{19} \\ p_{02} & p_{21} & p_{22} & p_{23} & p_{24} & p_{25} & p_{26} & p_{27} & p_{28} & p_{29} \\ p_{03} & p_{31} & p_{32} & p_{33} & p_{34} & p_{35} & p_{36} & p_{37} & p_{38} & p_{39} \end{bmatrix}_u$$ —model parameter matrix (4);

$$\mathbf{W}(\mathbf{X}_u(y,h)) = \begin{bmatrix} 1 & x_{1u}(y,h) & x_{2u}(y,h) & x_{3u}(y,h) & x_{1u}^2(y,h) & x_{2u}^2(y,h) \\ x_{3u}^2(y,h) & x_{1u}^3(y,h) & x_{2u}^3(y,h) & x_{3u}^3(y,h) \end{bmatrix}$$ —vector-function, where the arguments are the parameters of the state of sowing for the spatial coordinate ($y,h$): $x_{1u}$—the density of the sowing biomass, cwt·ha$^{-1}$; $x_{2u}$—the density of the sowing wet mass, cwt·ha$^{-1}$; $x_{3u}$—the density of the mass of ears, cwt·ha$^{-1}$.

A feature of the presented vector-matrix mathematical models (1)–(4) is that here the components of the vectors are not scalar values, but two-dimensional distributions of the corresponding biomass parameters in dynamic state models and reflection parameters in remote sensing models. This greatly complicates the modeling and estimation algorithms, as it leads to the need to introduce spatial cycles. The number of variables in such cycles depends on the method of dividing the total surface of the field into elementary sections. So, with the area of an elementary plot of 2 m$^2$, the number of cyclic variables will be 5000 per hectare of the field area. With a total field area under crops of 500 ha, the total number of elementary plots and cycles of the algorithm will be $2.5 \times 106$ units. Therefore, for large areas of crops (more than 1000 ha), it is advisable to use approximation schemes for modeling and estimation. The essence of such schemes lies in the fact that, first, the parameters of the sowing state averaged over the area of the field are modeled (estimated), which are then corrected along the field surface by means of a corrective model in the same way for the state of sowing before and after heading (omitting the indices of the phenological state of sowing).

A feature of the presented vector-matrix mathematical models (1)–(4) is that here the components of the vectors are not scalar quantities, but two-dimensional distributions of the corresponding biomass parameters in dynamic state models and reflection parameters in ERS models. This significantly complicates the modeling and estimation algorithms, and leads to the need to enter spatial cycles, where the number of variables depends on the method of dividing the total surface of the field into elementary sections. So, with an elementary plot area of 2–3 m$^2$, the number of cyclic variables will be 5000 per hectare of field area. With a total area of the field under sowing of 500 hectares, the total number of elementary plots and algorithm cycles will be $2.5 \times 10^6$ units. Therefore, for large areas of crops (more than 1000 hectares), it is advisable to use approximation modeling and estimation schemes. The essence of such schemes is that, first, the parameters of the seeding state averaged over the field are modeled (estimated), which are then corrected over the field surface by means of a corrective model in the same way for the seeding state before and after earing (omitting the indices of phenological phases of seeding)

$$\widehat{X}(t,y,h) = \widehat{X}(t) + \Delta\widehat{X}(y,h),$$
$$\Delta\widehat{X}(y,h) = K\Delta Z(y,h),$$
$$(5)$$

where: K—spatial corrector matrices for models (1) and (2), the parameters of which are estimated by forming an array of variations in the reflection parameters of remote sensing $\Delta Z(y,h)$ and estimating biomass parameters for 20–30 elementary plots.

### 2.2. Estimation Algorithm

To form estimates of biomass parameters averaged over the area of the field in the selected elementary plots, the following local estimation algorithm is used, built on the basis of models (2), (4) [2,8,11,12,16–20]

$$\dot{\widehat{X}}(t) = A\widehat{X}(t) + BV(t) + CF(t) + R(t)\frac{\partial W^T(P,\widehat{X})}{\partial\widehat{X}}K_z^{-1}(\widetilde{Z}(t) - \widehat{X}(t)),$$

$$\dot{R}(t) = R(t)A^T + AR(t) - R(t)\frac{\partial W^T(P,\widehat{X})}{\partial\widehat{X}}K_z^{-1}\frac{\partial W(P,\widehat{X})}{\partial\widehat{X}}P^TR(t),$$

(6)

where: $R(t)$—matrices of estimation errors, having the dimension corresponding to the vectors of the biomass parameters of the models (1), (4); $\widetilde{Z}(t)$—ERS data vector averaged over the field area.

In this case, the algorithm for constructing the spatial corrector (5) includes the following steps:

**Step1.** From the remote sensing data obtained over the entire area of the field, data on test plots are selected $Z(y,h) = Z(i)$, $i = 20...40$.

**Step2.** Remote sensing data for test areas are averaged $Z\frac{1}{20}\sum\limits_{i=1}^{20}Z(i)$.

**Step3.** For each test plot, the estimation algorithm (6) evaluates the state of the crop biomass $\widehat{X}(i)$, $i = 20...40$. The scores obtained are averaged $\widehat{X} = \frac{1}{20}\sum\limits_{i=1}^{20}\widehat{X}(i)$.

**Step4.** Determine local variations of remote sensing data $\Delta Z(i) = Z(i) - Z$ and local variations of estimates $\Delta\widehat{X}(i) = \widehat{X}(i) - \widehat{X}$, $i = 20...40$.

**Step5.** From the obtained variations, an array of data is formed, $\{\Delta\widehat{X}(i), \Delta Z(i)\}$ which evaluates the parameter matrix K of the spatial corrector (5).

This procedure is performed for the period of time preceding the heading of the crop and for the period from the beginning of the heading to the full ripening of the grain.

## 3. Results

### 3.1. Experimental Research Base

Approbation of the estimation problem was carried out at the experimental site of the Menkovsky branch of the Agrophysical Institute. Figure 1 shows a fragment of aerial photography of the experimental field of the Menkovsky branch with the selection of a field with spring wheat, which was the object of study. The fragment was obtained from a Geoscan 401 unmanned aerial vehicle equipped with a MikoSensRedH MX multispectral optical camera. Reflection parameters in the range from 430 to 950 nm were recorded by means of a multispectral camera over the entire field area under spring wheat.

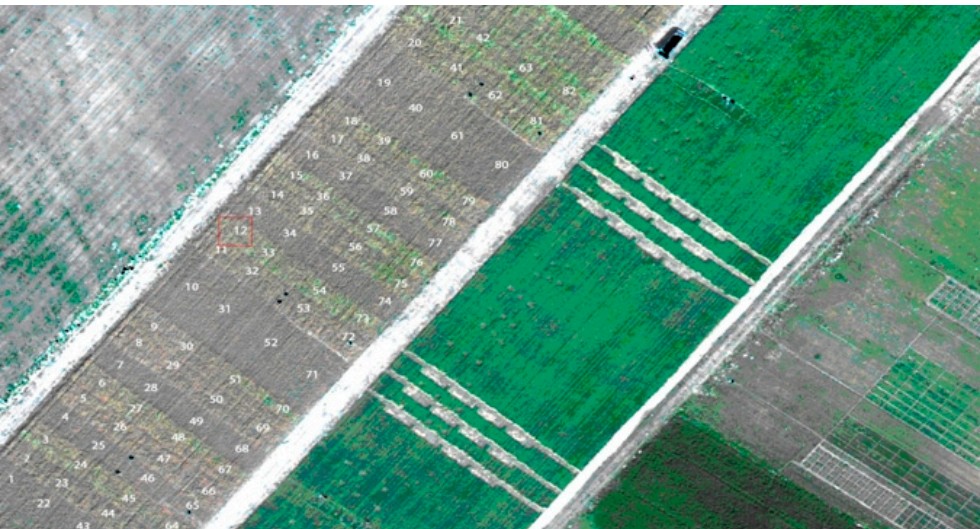

**Figure 1.** Image of an experimental field with sowing of spring wheat at the stage of milky-wax ripeness. Menkovsky branch of the Agrophysical Institute.

The field area was divided into 82 elementary sections, designated by numbers. Samples of crop biomass and soil were taken from 20such plots (nos. 20–40). The sampling of plants weighing 1 kg was carried out by cutting the biomass bypacking the samples in a sealed package. Soil samples were taken with a manual cylindrical probe from a soil layer 25 cm deep. Soil samples were placed in special sealed containers. The selection of such samples was carried out during the entire growing season.

The selected samples were analyzed in a laboratory to identify the physical and chemical parameters of the crop biomass and soil. The biomass structure was determined according to B.A. Dospekhov "Methodology of field experience", Moscow, Agropromizdat, 1985. The chemical composition was determined by the following methods and techniques: humidity—by the method of air-heat drying according to GOST 13586.5-2015, total nitrogen—by the photometric indophenol method for determining nitrogen (spectrophotometer PE-3000UF) according to GOST 13496.4-2019, phosphorus—by the photometric method (spectrophotometer PE-3000UF) after dry ashing according to GOST 26657-97, potassium was determined according to GOST 30504-97 by the flame photometric method after dry ashing, and calcium—by the complexometric method (determination of calcium in samples prepared by the dry ashing method) according to GOST 26570-95. (GOST—the state standard of Russia).

Simultaneously with sampling, surface remote sensing was carried out with a Hand-Held 2 Portable Spectroradiometer (Product Regulations Manager, ASD Inc.2555 55th St., Ste. 100, Boulder, CO 80301 USA). Based on this monitoring information, all mathematical models used in the estimation algorithm were adapted.

### 3.2. Results of Approbation of Models and Estimation Algorithms

In the time interval preceding heading, the reflection parameters were recorded in the visible and near-IR optical ranges. Figure 2 shows the dynamics of the sowing reflection parameters averaged over the field area in this time interval, and Figure 3 shows the estimates of the sowing biomass parameters obtained from remote sensing data without measuring these parameters in the field. Estimation errors for both parameters fit into the 10% tolerance field. Figures 4 and 5 present the same information for the growing season from the beginning of heading to grain ripening. Here, the reflection parameters in the green, red, and near-IR ranges of the optical spectrum have already been fixed, and the mass of ears (yield) has been estimated in the composition of the biomass. These structural changes did not affect the estimation accuracy.

Figures 4 and 5 present the same information for the growing season between the milky-wax and grain ripening phenophases. Here, the reflection parameters in the green, red, and near-IR ranges of the optical spectrum have already been recorded, and the weight of ears (yield) in the composition of the biomass was also estimated. These structural changes did not affect the accuracy of the estimates.

The same assessment procedure was carried out at 20 elementary plots, according to the results of which a database was formed, including two segments, before the beginning of earing and from the beginning of earing to the phase of grain ripening in the ear. The specified database is an information basis for setting up spatial correctors of estimates. Figures 6 and 7 show the graphs of setting the spatial correctors at both studied intervals of the growing season. As can be seen from the graphs, the tuning accuracy is quite high (within a 5% interval), which provides a reliable spatial correction of the biomass parameter estimates.

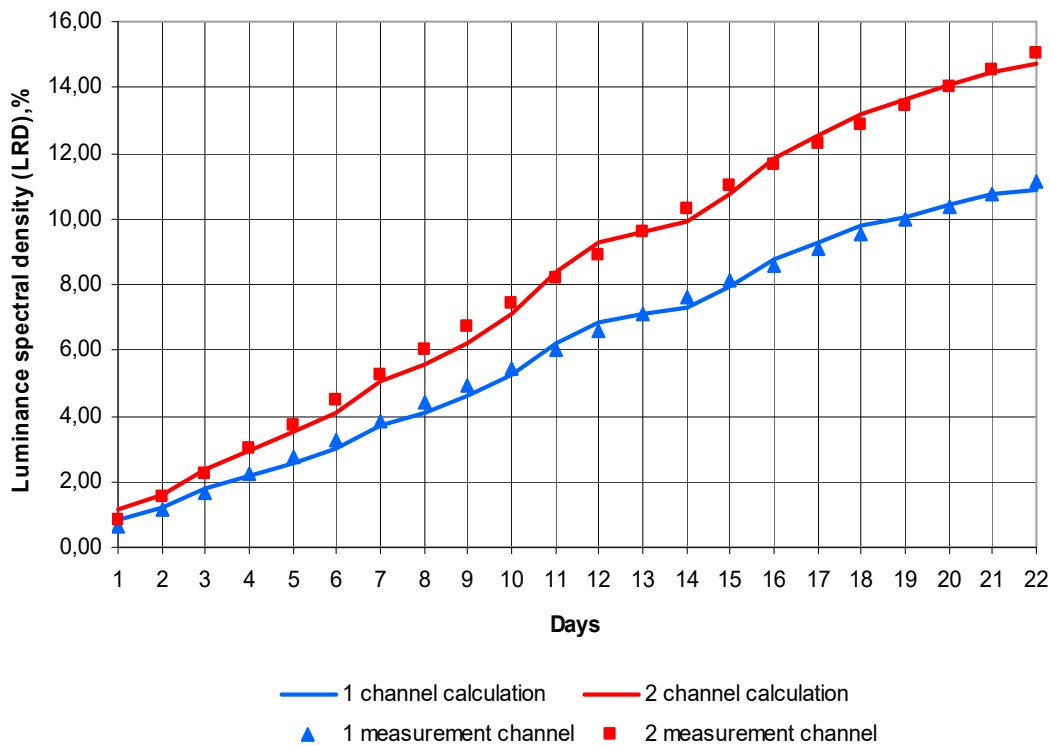

**Figure 2.** Dynamics of the field-averaged parameters of reflection of spring wheat sowing in the time interval preceding heading. 1 channel—red, 2 channel—near IR.

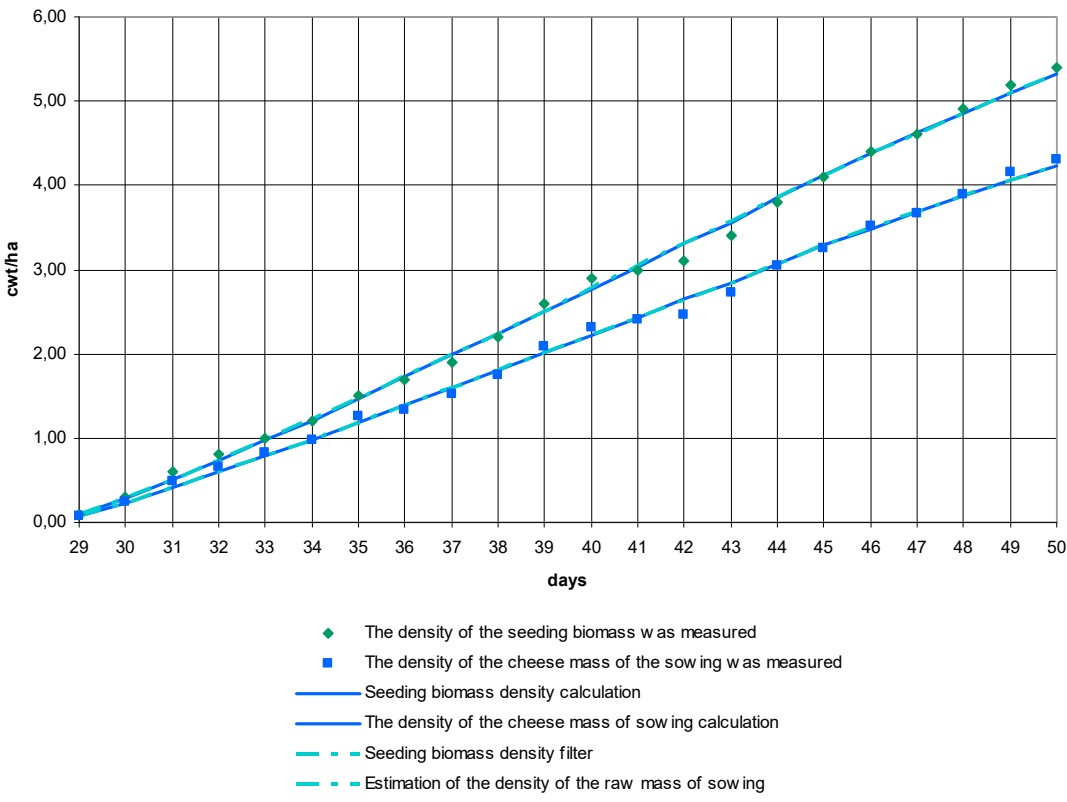

**Figure 3.** Estimates of the field-averaged parameters of the biomass of spring wheat sowing in the time interval preceding heading.

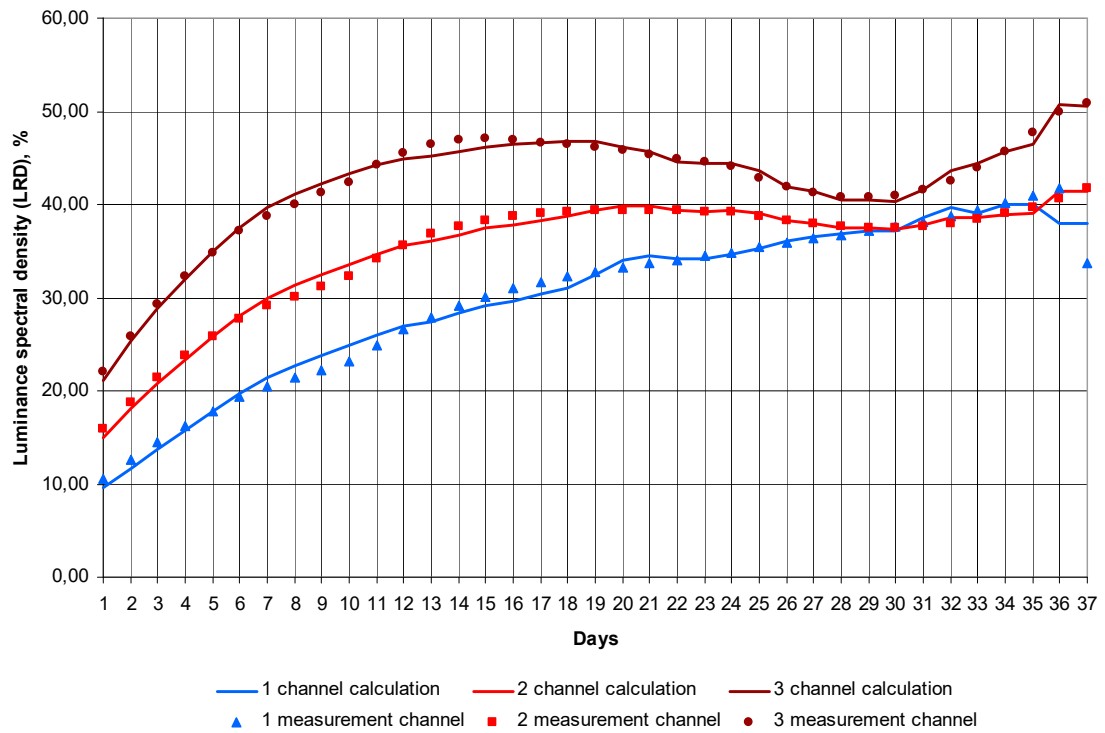

**Figure 4.** Dynamics of the reflection parameters of spring wheat sowing in the time interval from heading to grain ripening. 1 channel—green, 2 channel—red, 3 channel—near IR.

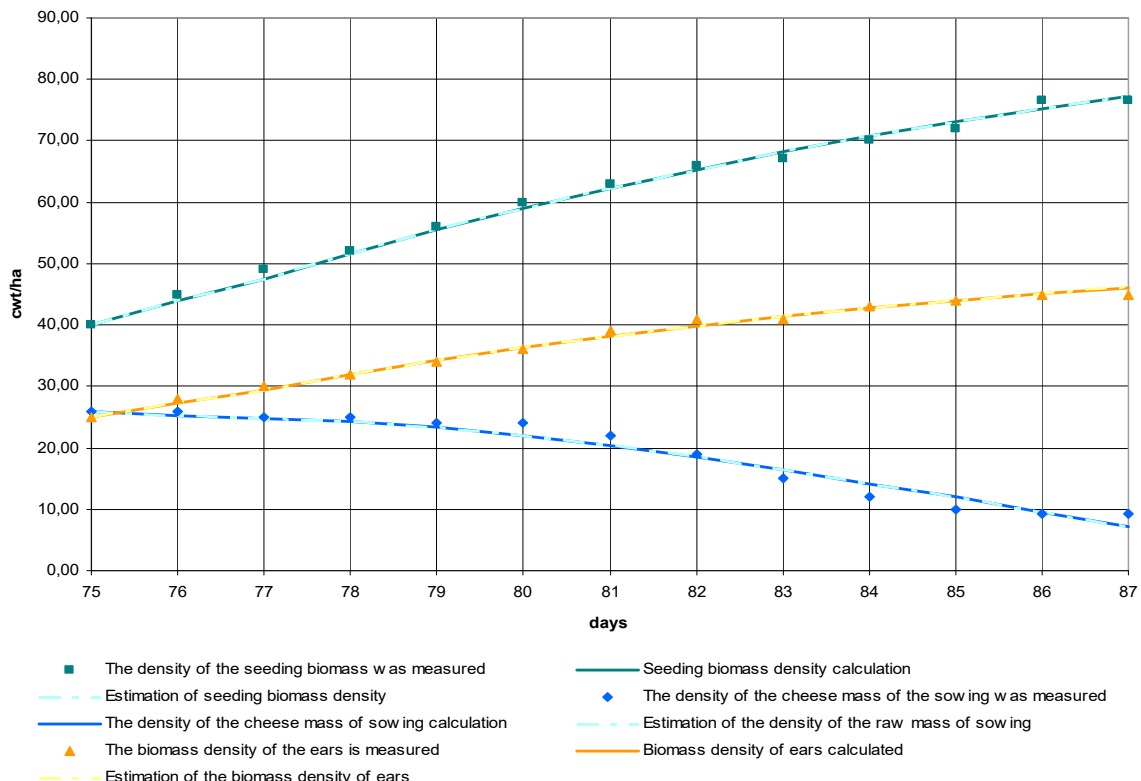

**Figure 5.** Estimates of the parameters of the biomass of spring wheat sowing in the time interval from heading to grain ripening.

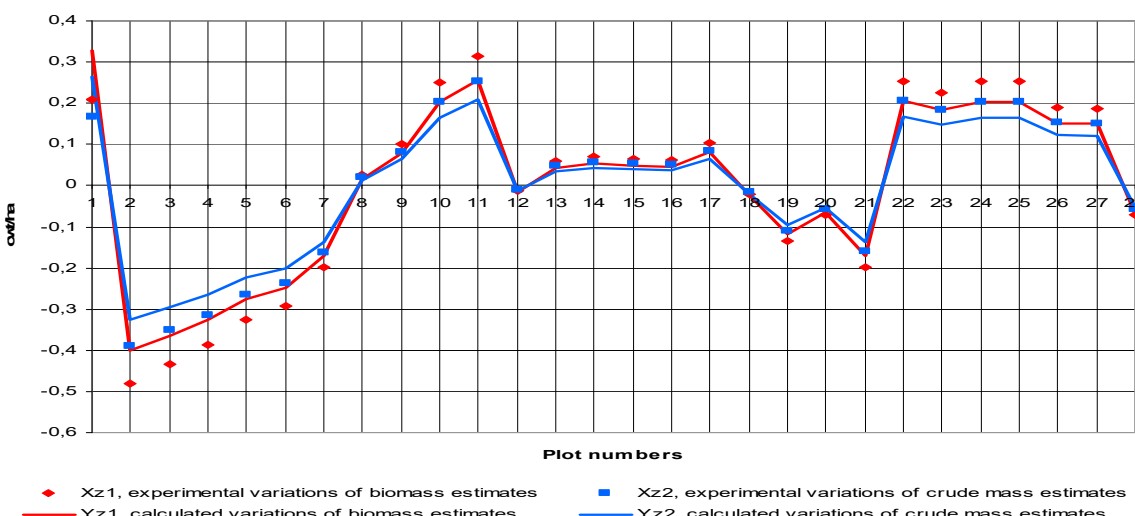

**Figure 6.** Adjustment of the spatial corrector of estimates of the biomass of spring wheat sowing in the time interval preceding heading.

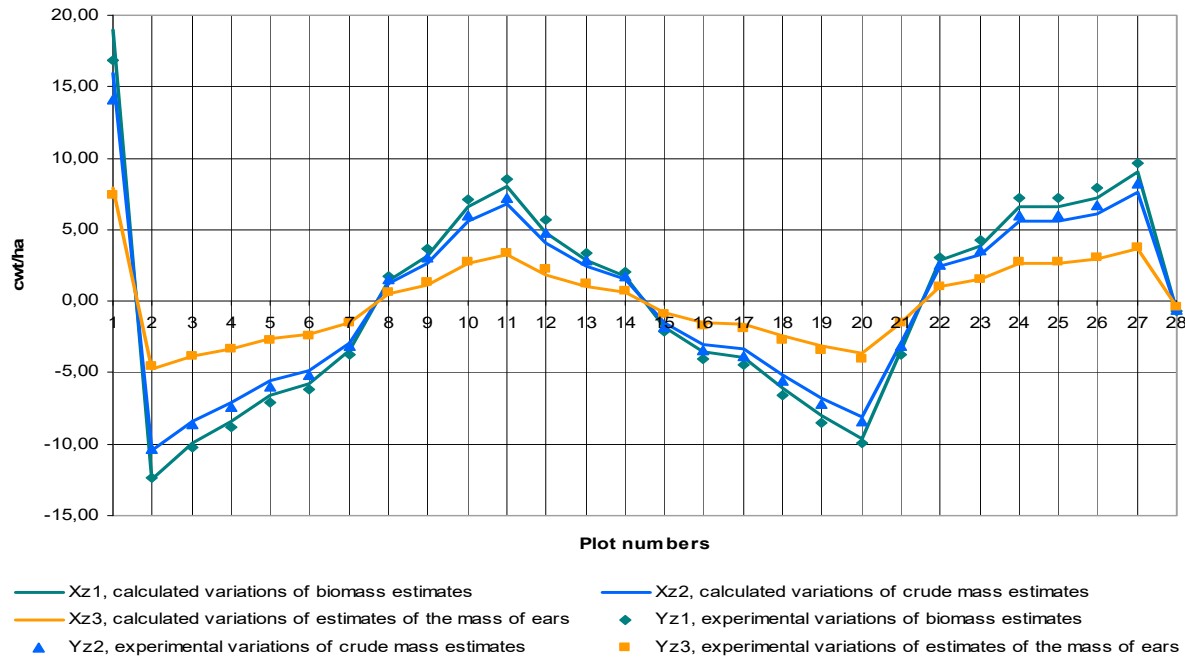

**Figure 7.** Adjustment of the spatial corrector of estimates of the biomass of spring wheat sowing in the time interval from heading to grain ripening.

Such estimates were constructed for the 70th day of the growing season, when the test field was flown around and ERS data were obtained in the used optical ranges. The distribution of these data over all 84 elementary areas is shown in Figures 8–13 show the distribution of estimates of the parameters of the seeding biomass. Such estimates can be obtained for any moment of the daily planting growing season.

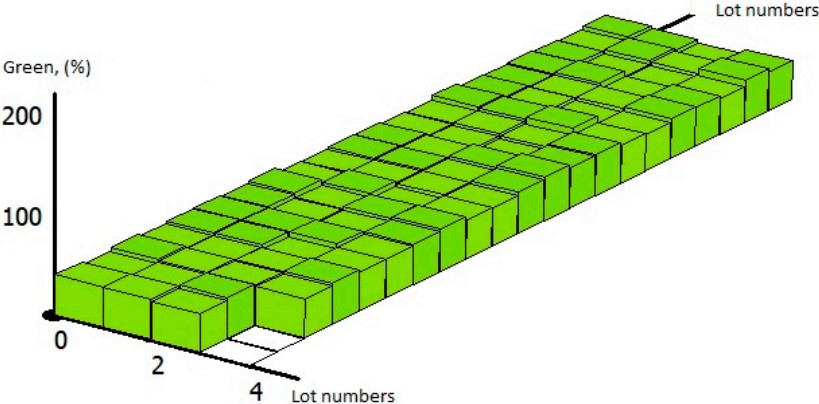

**Figure 8.** Distribution of the reflection parameters over elementary areas of the field with sowing of spring wheat in the "green" region of the spectrum.

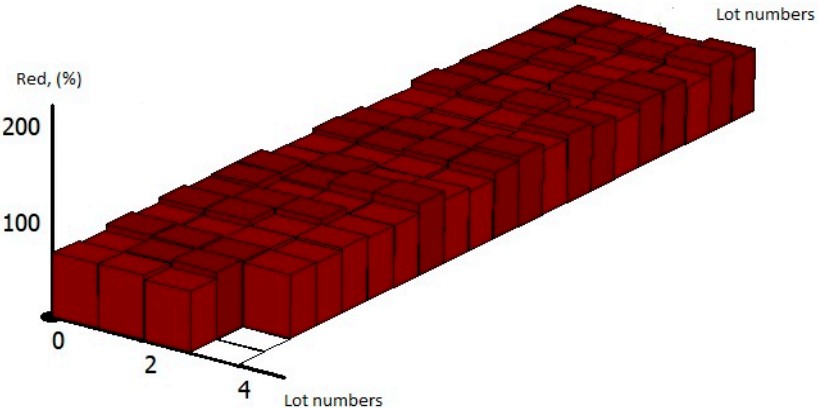

**Figure 9.** Distribution of the reflection parameters over elementary areas of the field with sowing of spring wheat in the "red" region of the spectrum.

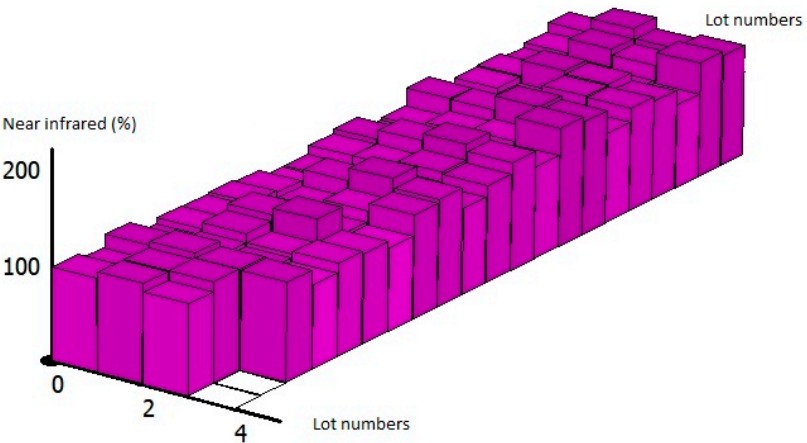

**Figure 10.** Distribution of the reflection parameters over elementary areas of the field with sowing of spring wheat in the "near infrared" region of the spectrum.

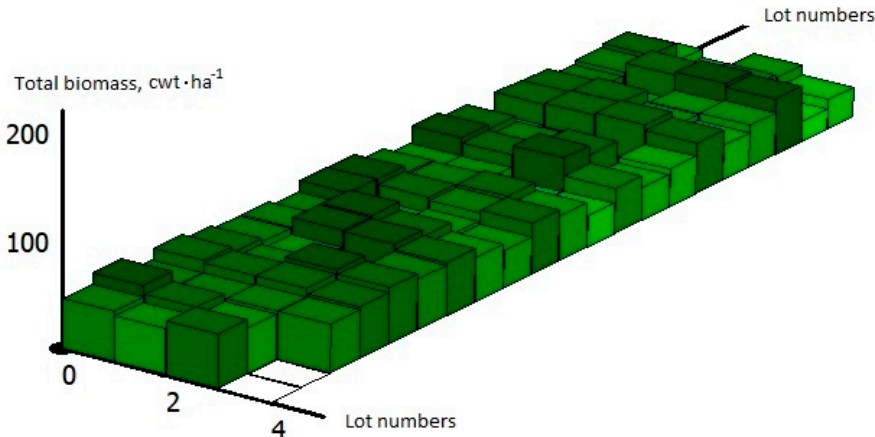

**Figure 11.** Distribution of the estimates of the total biomass of spring wheat sowing by elementary plots of the field.

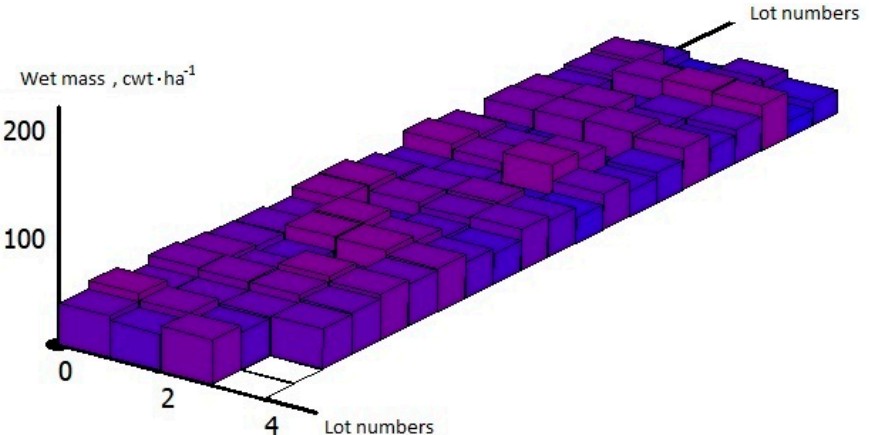

**Figure 12.** Distribution of the estimates of the wet weight of spring wheat sowing by elementary plots of the field.

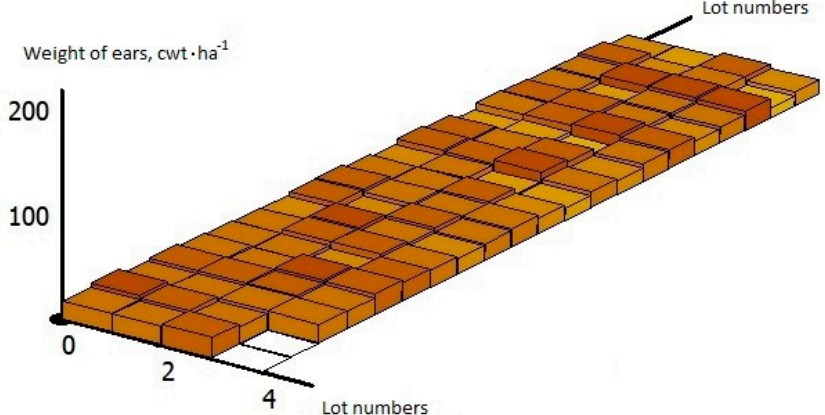

**Figure 13.** Distribution of the estimates of the weight of ears of spring wheat sowing by elementary areas of the field.

### 3.3. The Discussion of the Results

The results of approbation of the proposed estimation methodology show that the user of the software product developed on its basis can have at his disposal estimates of the parameters of the sowing biomass in each elementary plot of the field with an area of 2–4 m². First of all, such information is in demand in precision farming systems (TK),

in which automated technological machines operate, where elementary sowing areas can be serviced by working bodies independently of each other [17]. This makes it possible to realize high accuracy of control over the state of crops in conditions of large spatial heterogeneity of the agricultural field. Such inhomogeneities are due to the influence of soil heterogeneity, uneven sowing of seeds, differences in the rate of development of individual groups of plants, the presence of a field microrelief, and other reasons. In addition to management tasks in TK systems, such information is in demand in monitoring systems. It is designed to analyze and predict the state of crops, both over the entire area of the field, and in its individual zones and elementary sections. With its help, it is possible to accurately determine areas of low productivity, soil degradation, and stress conditions of crops.

The proposed methodology can be applied to the sowing of other agricultural crops. For this, it is necessary to clarify the structure of the mathematical models used, without significant changes in the algorithms and programs used. The main direction of the development of this methodology is to improve the quality of the used mathematical models of the parameters of the state of agricultural crops. This can be achieved by introducing 3D models of the agricultural field landscape. The introduction of such models will allow taking into account the effect of relief on the distribution of moisture and fertilizers and thereby significantly increase the accuracy of spatial modeling of the parameters of the state of the biomass of the studied crops.

### 4. Conclusions

A new methodology and algorithm for estimating the parameters of the crop biomass based on Earth remote sensing (ERS) data are proposed. It is based on the classical approach to the estimation problem based on the use of mathematical models of the estimated parameters and their relationship with remote sensing data. A distinctive feature of the methodology is the transition to models with spatial variables and accounting for the phenological phases of cereal sowing. At the same time, to simplify the computational procedures associated with the presence of spatial variables, an approximation approach based on the use of linear spatial correctors, the number of which is equal to the number of used mathematical models of the parameters of the state of the crop biomass, is proposed. Moreover, the construction of spatial correctors is carried out by repeatedly solving the estimation problem in separate elementary sections, highlighting the spatial variations of the remote sensing data and estimates, according to which the parameters of the spatial correctors are adjusted.

**Funding:** The study was supported by the grant of the Russian Foundation for Basic Research No. 16-07-00925/16 "Development of theoretical foundations and software and hardware for assessing the state of crops and soil environment according to remote sensing data of the Earth".

**Data Availability Statement:** Data confirming the presented results can be found at: Laboratory of Information and Measurement Systems 130, Agrophysical Research Institute, 195220, St. Petersburg, GrazhdanskyProspekt, 14, labit@yandex.ru.

**Conflicts of Interest:** The author declares no conflict of interest.

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
