# Peer review of "Estimation of Parameters of Biomass State of Sowing Spring Wheat"

_remotesensing, doi:10.3390/rs14061388_

Round 1

Reviewer 1 Report

Attached is a document addressed primarily to Authors but also for the consideration of Editors

Author Response

In accordance with the comments and suggestions of the reviewers, the text of the article was revised almost in its entirety. The following changes and additions have been made:

  1. Completely revised text of the abstract. It is made more concise, it clearly highlights the purpose of the work and its connection with the results obtained.
  2. The introduction has been revised taking into account the involvement of new sources, the connection of the proposed estimation method with previous research results over a long period of analysis is highlighted.
  3. The reviewer's suggestion on the presentation of the formulas of the mathematical models used has been taken into account. They are presented in a single format, which makes them easier to read and understand.
  4. Methods and means of sampling in field experiments and laboratory analyzes are also presented.
  5. Revised conclusions with an emphasis on the results obtained and their advantages compared to the known results.
  6. The text discloses all the technical means of remote sensing used.
  7. An algorithm for constructing a spatial corrector is presented, which simplifies the modeling and estimation procedure due to the presence of spatial variables.
  8. In lines 105-112 - an explanation is given for the units of mass measurement adopted in Russia.
  9. Lines 184-186 provide an explanation of the samples taken and the methods of laboratory analysis used.
  10. Added all definitions for wheat biomass parameters.
  11. In line 234, the repeated words 'with sowing' have been removed from the heading of figure 8.

Line 237 removed the repeated words 'with sowing' from the title of figure 9.

On line 240, the repeated words "with seed" have been removed from the heading of figure 10.

  1. Improved the English language of the text of the article.

Author:

Ilya Mikhailenko

Reviewer 2 Report

The work entitled: Estimation of Parameters of Biomass State of Sowing Spring Wheat, is useful for the development of research in the field of sustainable agriculture as it provides the basis for developing new methods of investigation of the areas affected by crops. While appreciating the work done, the paper requires a significant revision by the authors.

  1. Specifically, it is advisable to streamline the text of the abstrac, making it more concise and highlighting the purpose of the work in relation to the results obtained.
  2. It is requested to improve the introduction by referring to more bibliographic sources
  3. In order to make reading easier and clearer the work developed in the description of materials and methods, it is necessary to standardize the equations using a single expression applied to all the equations reported in the text. It is required to rewrite the equations using the same format.
  4. in the lines 105-112 it is necessary to better express the units of measurement paying attention to the use of special characters.
  5. About the sentece in lines 184-186 "The selected samples were analyzed in a laboratory way to identify the physical and chemical parameters of the biomass of crops and soil." Specify the nature of selected samples, the kind of characterization analisys, with reference to standard methods used for the determination of physical and chemical characterization and report the results about these parameters. With reference to this sentence, explains the importance to use these parameters for the estimation of biomass state system.
  6. It is suggested to improve the text of the conclusions by referring to the results and the advantages obtained from the study carried out. 

Author Response

(The authors gave the same response as above.)

Reviewer 3 Report

The present study proposed a new methodology and algorithm for estimating the parameters of the biomass of crops sowing based on Earth remote sensing (ERS) data. The distinctive feature of the methodology is the transition to models with spatial variables and accounting for the phenological phases of cereal sowing, which the  construction of spatial correctors is carried out by repeatedly solving the estimation problem in separate elementary sections, highlighting the spatial variations of the remote sensing data and estimates, according to which the parameters of the spatial correctors are adjusted. This research make some novelty in methodology for spring wheat biomass estimation. The mechanism behind the relationships was explained and discussed. The manuscript falls well within the scope of this journal. However, this paper is not well structured, the main part of the methods and analysis seems to need improved. I have some concerns about the abstract, methods and results presentations. I think it needs major modifications before it can be accepted for publication in this journal.

  1. More detail about the spring wheat sowing experiment should be introduced in the   section Materials and methods.
  2. Dose all the Earth remote sensing (ERS) data used in this study collected by UAV mounted sensor? Please introduce the detail information about the sensors and add more information about the UAV flight experiment.
  3. Please add all the definition of the wheat parameters used in study, such as sowing biomass, total biomass, wet weight, weight of ears.
  4. Line 154 -Line 174, suggest the author use a schematic or graph to the descript how to simplify the computational procedures associated with the presence of spatial variables. It is helpful for readers to understand the authors methodology and algorithm.
  5. Line 184, please indicate the twenty biomass sample sites in Figure 1.
  6. Line 230, seeding biomass should be sowing biomass.
  7. Line 234, please remove the repeated words ’with sowing’ from title Figure 8.
  8. Line 237, please remove the repeated words ’with sowing’ from title Figure 9.
  9. Line 240, please remove the repeated words ’with sowing’ from title Figure 10.

Author Response

(The authors gave the same response as above.)

Round 2

Reviewer 2 Report

Numerous changes have been made and the paper now has a greater scientific soundness.
The suggestions were accepted.
Small changes of form are suggested and to reduce the number of self-citations.

Author Response

In accordance with the comments and suggestions of the reviewers, the following changes and additions were made in the text of the article:

  1. Sections 2 and 3 of the article have been reorganized. Additional section subheadings have been introduced to make the article easier to read and understand.
  2. In line 111, repeated words have been removed.
  3. In lines 375-418, the numbers of all references are checked.
  4. An English teacher looked at the article. No major errors were found in the text. We will eliminate minor spelling remarks when working on the layout of the article.

Author:

Ilya Mikhailenko

Reviewer 3 Report

This manuscript has been improved a lot by the authors. The scientific soundness of this manuscript have been increased significantly. But the study results are still necessary presented in a more efficient and effective way.

The author just introduced the theory and methodology of the study in section 2. All materials about this study was showed in the section 3 part. I suggest the author reorganize those two sections. More sub-titles should be added to the section Materials and methods, such as 2.1 Theory methods and 2.2 Experiment and Material. The section 3 is the Results part. I also suggest the author consider adding subheadings to make the study results presented in a more efficient and effective way.

Some minor modifications are needed for this manuscripts.  

Line 111, please remove the repeated words ’Materials and methods’ from title.

Line 375-418, please check the number of all the reference.

Author Response

(The authors gave the same response as above.)
